# (-)-Epigallocatechin-3-Gallate Attenuates the Adverse Reactions Triggered by Selenium Nanoparticles without Compromising Their Suppressing Effect on Peritoneal Carcinomatosis in Mice Bearing Hepatocarcinoma 22 Cells

**DOI:** 10.3390/molecules28093904

**Published:** 2023-05-05

**Authors:** Qiuyan Ban, Wenjing Chi, Xiaoxiao Wang, Shiqiong Wang, Dan Hai, Guangshan Zhao, Qiuyan Zhao, Daniel Granato, Xianqing Huang

**Affiliations:** 1College of Horticulture, Henan Agricultural University, Zhengzhou 450002, China; banqiuyan717@163.com (Q.B.); cwj15698245770@163.com (W.C.); 2College of Food Science & Technology, Henan Agricultural University, Zhengzhou 450002, China; dora8261@163.com (X.W.); shiqiongwang@163.com (S.W.); danhai71@henau.edu.cn (D.H.); 15003847166@163.com (Q.Z.); 3College of Life Science and Technology, Jinan University, Guangzhou 510632, China; 4Department of Biological Sciences, Faculty of Science and Engineering, University of Limerick, V94 T9PX Limerick, Ireland; daniel.granato@ul.ie

**Keywords:** (-)-epigallocatechin-3-gallate, selenium nanoparticles, chemotherapy, chemotherapy-induced diarrhea, antioxidant defense

## Abstract

Increasing evidence shows that selenium and polyphenols are two types of the most reported compounds in tumor chemoprevention due to their remarkable antitumor activity and high safety profile. The cross-talk between polyphenols and selenium is a hot research topic, and the combination of polyphenols and selenium is a valuable strategy for fighting cancer. The current work investigated the combination anti-peritoneal carcinomatosis (PC) effect of selenium nanoparticles (SeNPs) and green tea (*Camellia sinensis*) polyphenol (-)-epigallocatechin-3-gallate (EGCG) in mice bearing murine hepatocarcinoma 22 (H22) cells. Results showed that SeNPs alone significantly inhibited cancer cell proliferation and extended the survival time of mice bearing H22 cells. Still, the potential therapeutic efficacy is accompanied by an approximately eighty percent diarrhea rate. When EGCG was combined with SeNPs, EGCG did not affect the tumor proliferation inhibition effect but eliminated diarrhea triggered by SeNPs. In addition, both the intracellular selectively accumulated EGCG without killing effect on cancer cells and the enhanced antioxidant enzyme levels in ascites after EGCG was delivered alone by intraperitoneal injection indicated that H22 cells were insensitive to EGCG. Moreover, EGCG could prevent SeNP-caused systemic oxidative damage by enhancing serum superoxide dismutase, glutathione, and glutathione peroxidase levels in healthy mice. Overall, we found that H22 cells are insensitive to EGCG, but combining EGCG with SeNPs could protect against SeNP-triggered diarrhea without compromising the suppressing efficacy of SeNPs on PC in mice bearing H22 cells and attenuate SeNP-caused systemic toxicity in healthy mice. These results suggest that EGCG could be employed as a promising candidate for preventing the adverse reactions of chemotherapy including chemotherapy-induced diarrhea and systemic toxicity in cancer individuals.

## 1. Introduction

Nowadays, natural products are employed as chemoprevention drugs. Polyphenols, alkaloids, flavonoids, polysaccharides, and carotenoids are biologically active and have a wide spectrum of effects [1,2,3]. (-)-Epigallocatechin-3-gallate (EGCG), a natural substance mainly originated from the tea plant (*Camellia sinensis*), has many advantages over synthetic drugs due to its high safety profile, low cost, and easy availability and is of great interest. (-)-Epigallocatechin-3-gallate (EGCG), the most abundant catechin with the highest redox activity among green tea polyphenols, has been reported to have many health benefits, including anticancer effects [4]. EGCG is known as a typical antioxidant with pro-oxidation capacity. It was demonstrated that EGCG is easily auto-oxidized in the medium of cultured cancer cells to produce H2O2, superoxide, or other types of reactive oxygen species (ROS) [5,6], thus inducing cell death; the addition of superoxide dismutase (SOD) or catalase attenuated these actions of EGCG [7]. Similarly, SOD or catalase abolished EGCG-induced ROS production, DNA damage, and cell apoptosis in human lung cancer H1299 cells and in xenograft tumors in mice [8]. Thus, the pro-oxidant activity of EGCG plays an important role in stimulating ROS production, oxidative damage, and cell death. On the other hand, a daily intake of three or six green tea polyphenol tablets containing 237 or 474 mg polyphenols effectively reduces oxidative stress induced by hepatic arterial infusion chemotherapy (with cis-platinum (II) diamine dichloride or 5-fluorouracil) in hepatocellular carcinoma or metastatic liver cancer patients [9]. EGCG protects against the damage induced by the chemotherapy drug cis-platinum (II) diamine dichloride and gamma irradiation in normal salivary gland cells [10]. EGCG enhances the radiation sensitivity in HCT-116 colorectal cancer cells via increasing nuclear factor (erythroid-derived 2)-like 2 (Nrf2) activation and nuclear translocation [11].

Selenium (Se) is known as an essential trace element with several beneficial effects. At healthy doses, Se plays a crucial role in scavenging reactive oxygen species (ROS) and is an essential constituent of antioxidant enzymes, including thioredoxin reductase (TrxR) and glutathione peroxidase (GPx) [12,13]. However, at high doses, certain types of Se have potent cell-killing capacity by stimulating the formation of ROS [14,15,16]. Selenium nanoparticles (SeNPs) have lower systemic toxicity compared to selenomethionine or methylselenocysteine because of lower levels of Se accumulation in tissues following oral administration of SeNPs in mice [15,16,17], and SeNPs have higher redox activity in stimulating ROS production compared to selenite [18], which is known as one of the most redox-active Se forms [19,20]. Previous studies found that SeNPs showed high safety and impressive antitumor activity in mice in preventing or treating peritoneal carcinomatosis (PC) [18,21].

Cancer has been one of the leading causes of mortality worldwide for a long time. Well-directed chemotherapy and radiotherapy are the current clinical-based approaches for the treatment of most cancers [22,23]. However, these treatments often cause various adverse reactions and lower the overall quality of life of patients. The current guidelines for protecting against these side effects are insufficient in clinics [24]. Thus, combining clinical anticancer drugs with compounds with anticancer activity to access better efficacy without noticeable side effects represents a paradigm shift for cancer therapy. Increasing evidence shows that selenium and polyphenols are two types of the most reported compounds in tumor chemoprevention due to their remarkable antitumor activity and high safety profile [25,26,27]. The cross-talk between polyphenols and selenium is a hot topic and has high novelty. However, this topic has not been well investigated, especially regarding the combined antitumor effect in vivo. Given the highly effective inhibition effect of SeNPs on PC [18,21], in the current work, we investigated the combined effect of EGCG and SeNPs on PC in mice bearing murine hepatocarcinoma 22 (H22) cells to realize the influences of EGCG on the anticancer activity and adverse reactions of SeNPs in fighting cancer.

## 2. Results and Discussion

### 2.1. Effects of SeNPs on Peritoneal Carcinomatosis in Mice Bearing H22 Cells

PC has the potential to disseminate and grow in the peritoneal cavity, thus leading to tumor recurrence and the formation of malignant ascites, numerous small tumor nodules, or various sizes of tumor masses [28]. After H22 cells were injected into the abdominal cavity, the body weight of mice increased abnormally because of the quick proliferation of H22 cells and the accumulation of ascites [21]. Firstly, the therapeutic efficacy of SeNPs alone was inspected. H22 model mice were i.p. injected with saline (control group) or 3 mg/kg SeNPs (treatment group). SeNPs remarkably inhibited the proliferation of H22 cells in the abdominal cavity of mice at 48 h post-treatment (Figure 1A), which was also indicated by the reduced body weight compared to the control (Appendix A). Still, SeNPs caused a diarrhea rate of eighty-three percent at 24 h post-treatment (Figure 1B). In the long-term survival experiment, SeNPs significantly prolonged the survival time of mice (Figure 1C). However, widespread diarrhea was again observed in mice (Figure 1D). The antitumor mechanism of SeNPs mainly involved selenium selectively accumulated in cancer cells to levels as high as 40 times compared to the control (Figure 2A), but selenium only increased no more than 1 time in the small intestine or colon (Figure 2B). In addition, SeNPs upregulated caspase 8 and 9 activities (Figure 2C), downregulated Bcl-2 and caspase 8 protein expression levels and the ratio of Bcl-2/Bax (Figure 2D), and caused drastic protein degradation (Figure 2E) in H22 cells. Evidence shows that selenium induces apoptosis through superoxide-mediated activation of mitochondrial pathways [18,29,30], indicating that the accumulation of superoxide in cancer cells plays a crucial role in modulating oxidative stress levels and cell death caused by selenium. SeNPs have higher redox activity in inducing ROS production than selenite, which has been validated in Grx and Trx pure enzyme systems [18]. EGCG is known as an antioxidant with the property of pro-oxidation dependent on the specific environment and dose level. Herein, we explored the effect of SeNPs on inducing ROS production in an H22 cell suspension and the influence of EGCG on SeNP-induced ROS production in the system. This will help to understand the interaction between EGCG and SeNPs in vitro, thus facilitating the following research on the combined effect of EGCG and SeNPs in vivo.

### 2.2. EGCG Enhances the Production of SeNP-Induced ROS in H22 Cell Suspension

Both SeNPs and EGCG induced ROS production in a dose-dependent manner in the H22 cell suspension (Figure 3A,B). EGCG dose-dependently enhanced SeNP-triggered ROS levels (Figure 3C). To verify the synergistic effect of EGCG and SeNPs on ROS production, we inspected the effect of EGCG-stabilized SeNPs (E-SeNPs) on ROS production in H22 cell suspension. E-SeNPs were prepared in the redox system of GSH and sodium selenite in the presence of EGCG as disperser at pH 8.0; the molar ratio of EGCG/selenium or GSH/selenium was set at 4, and nano-sized EGCG and selenium were thus obtained [31]. Results showed that E-SeNPs can stimulate the production of ROS (Figure 3D) and show higher properties than SeNPs or EGCG alone at the same dose (Figure 3E), which was consistent with the result of combining EGCG and SeNPs described above. However, the strong acidic pH condition leads to massive aggregation of E-SeNPs by protonation in hydrochloric acid from gastric juice, thus reducing the bioavailability of E-SeNPs compared to size-equivalent SeNPs in regulating hepatic and renal GPx activity and hepatic Se levels by oral administration in mice [31]. EGCG enhanced SeNP-induced ROS production (Figure 3B), and E-SeNPs had higher activity in promoting ROS production than EGCG or SeNPs alone in the H22 cell suspension (Figure 3E); all these results suggested that the combination of EGCG and SeNPs is a potential approach for treating cancer by increasing ROS levels. Therefore, we next examined the combined anti-peritoneal carcinomatosis effect of EGCG and SeNPs in mice bearing H22 cells.

### 2.3. EGCG Reduces the Diarrhea Proportion Caused by SeNPs without Compromising Their Suppressing Efficacy on Peritoneal Carcinomatosis in Mice

H22 model mice were i.p. injected with SeNPs (4 mg/kg) or SeNPs (4 mg/kg) plus EGCG (55 mg/kg) once. Viable cells in the abdominal cavity were collected at 48 h post-treatment. Results showed that EGCG did not affect the killing effect of SeNPs on H22 cells (Figure 4A) but eliminated the diarrhea triggered by SeNPs (Figure 4B). Chemotherapy-induced diarrhea (CID) is known as one of the main drawbacks for cancer patients, the CID incidence has been reported to be as high as 50–80% of treated individuals [32,33]. Serious CID causes significant mortality. Currently, the strategies to overcome CID mainly include dose delay, dose reduction, or complete chemotherapy termination, thus broadly limiting the therapeutic efficacy [32,33,34]. Therefore, agents with a protective property are required to reduce CID morbidity and to improve the quality of life and clinical outcomes among cancer patients [32,33]. It has been demonstrated that SeNPs have higher redox activity compared to sodium selenite [18], a lower toxicity profile than methyl selenocysteine and selenomethionine [35,36], and a strong suppression effect on H22 cells in the abdominal cavity of mice [21]. Thus, SeNPs should be an optimized selenium species. However, they caused a diarrhea rate of approximately eighty percent in treating PC in mice (Figure 1B,D and Figure 4B). EGCG protected against diarrhea triggered by SeNPs without reducing their therapeutic efficacy on PC (Figure 4A,B), partly indicating that EGCG is a potential candidate for preventing CID. Literature shows that EGCG supplementation enhanced the gene expression of Nrf2 in the small intestine in Arbour Acres broilers chickens, thus alleviating the oxidative gut injury and inflammatory response of heat-stressed broilers [37], and EGCG increased Nrf2 activation and nuclear translocation when combined with radiation in treating HCT-116 colorectal cancer cells [11]. Since Nrf2 plays an important role in defending against oxidative damage [38,39], the property of EGCG activating the gastrointestinal Nrf2 pathway may involve the protective effect of EGCG on SeNP-triggered diarrhea (Figure 1A,B and Figure 4B) or potential injury induced by an elevated selenium level in the small intestine or colon (Figure 2B).

EGCG possesses a definite anticancer effect of inhibiting tumorigenesis in several animal models for different cancers and human cancer stem cells [4,40,41]. However, to date, the anticancer effect of EGCG on PC has not been reported. It was unexplainable that the toxic dose of EGCG (55 mg/kg, i.p.) [38,42] did not affect the inhibiting effect on H22 cells in the abdominal cavity in mice when combined with SeNPs, so next we explored the killing effect of EGCG alone on PC in mice bearing H22 cells by multiple intraperitoneal administration of 55 mg of EGCG per kg body weight.

### 2.4. Effect of EGCG on Peritoneal Carcinomatosis in Mice Bearing H22 Cells

EGCG (55 mg/kg) was delivered to H22 cells in the abdominal cavity of mice by i.p. injection for seven consecutive days. Surprisingly, EGCG did not have a suppressing effect on H22 cells, as indicated by the equivalent body weight gain of mice compared to the control (Figure 5A). However, after a single i.p. injection of 55 mg of EGCG per kg body weight, the intracellular EGCG level reached 0.933 nmol/mg protein (Figure 5B), but only 0.011 or 0.006 nmol/mg protein in the liver or small intestine (Figure 5B), respectively, at 1 h after the injection. EGCG selectively accumulated in H22 cells to levels almost two orders of magnitude higher than those in the liver or small intestine but without cytotoxicity, suggesting the H22 cells were insensitive or resistant to EGCG. Then a moderate malignant tumor model (2 million H22 cells were inoculated) was employed to study the preventive effect of EGCG on PC. After treatment for seven consecutive days, EGCG (50 mg/kg) did not inhibit the proliferation of H22 cells (Figure 5C) but significantly increased the activity of antioxidant enzymes, including GPx and GR in ascites (Figure 5D). 

### 2.5. EGCG Protects against SeNP-Triggered Systemic Toxicity via Restoring Antioxidant Defense in Healthy Mice

In the EGCG and SeNP combination group, EGCG (30 or 60 mg/kg) was delivered to mice before SeNPs for 24 h by i.p. injection. EGCG only moderately suppressed the body weight of mice on day 1 after EGCG administration (Figure 6A). SeNP administration caused liver toxicity, indicated by the significantly increased serum ALT and AST activities (Figure 6B,C). It decreased serum SOD, GSH, and GPx levels (Figure 6G–I), but EGCG prevented SeNP-caused body weight loss compared to SeNP administration alone at day 2 (Figure 6A) by restoring serum SOD, GSH, and GPx levels (Figure 6G–I). SeNP treatment and EGCG combined with SeNP treatment did not affect serum Cr, BUN, and LDH levels (Figure 6D–F). Moreover, the property of EGCG, in a pharmacologic or toxic dose, in activating antioxidant enzymes was also confirmed by the increased hepatic enzyme activity levels, including TrxR and Trx, after 5 or 7 consecutive days of treatment with EGCG in healthy mice (Figure 7). 

EGCG treatment significantly decreased the major antioxidant enzymes and the Nrf2-target genes at the lethal dose but enhanced the antioxidant defense system at the pharmacological or toxic dose [38,42]. Thereby, the lack of an inhibiting effect of EGCG on H22 cells may be due to the activated extracellular Grx antioxidant defense system, as indicated by the increased activity levels of enzymes, including GPx and GR in ascites (Figure 5D). TrxR and Trx are the core components of the Trx system and play essential roles in mediating cellular redox signaling pathways. Overexpressed TrxR showed pro-survival effects and enhanced tumor resistance to therapeutic modalities in many cancer cell lines [43,44]. However, increasing evidence indicates that TrxR or Trx has been recognized as a key modulator of tumor development; hence, targeting TrxR or Trx is a potential strategy for cancer therapy [45,46,47]. For instance, TrxR inhibitors have been proven to be promising drugs for cancer chemotherapy [48,49]. Since SeNPs can promote ROS production via the Grx and Trx systems [18] without compromising their key components, including antioxidant enzymes [16,18], combining SeNPs with Grx or Trx system activators may enhance the anticancer effect because SeNPs can persistently rely on these uncompromised and even enhanced systems to produce abundant ROS to suppress cancer cells [18]. In the present study, we investigated the co-effect of EGCG with SeNPs in fighting cancer. Considering the crucial role of Grx and Trx system activation in enhancing the anticancer effect of SeNPs, we conjecture that many EGCG-like antioxidants may improve the anticancer effect of SeNPs because of their unique performance in activating the Grx or Trx system. For instance, quercetin, astragalin, and rutinum can increase the activity of GPx, GR, and GST [50,51,52]. Curcumin can enhance the activity of GPx [53]. Catechins, including epigallocatechin and catechin, possess the capacity to activate TrxR1 and GPx1 [54,55].

As we know, stability and bioavailability limit the application of EGCG in clinical settings. A nanodelivery system has numerous advantages such as stability, biocompatibility, cellular uptake, and targeted drug delivery for enhancing its efficacy in the treatment of cancer and other diseases. Indeed, EGCG nanoparticles and SeNPs have been proven to show higher efficacy and safety than EGCG [56,57] or selenite [15,16,17,18]. Nanoparticle-mediated delivery enhances the bioavailability, eliminates unwanted toxicity of chemopreventive agents, and enhances the outcome of chemoprevention. Nanochemoprevention is a novel strategy for fighting diseases, although the nanosafety issue remains a controversial topic worldwide.

## 3. Materials and Methods

### 3.1. Chemicals and Materials

EGCG (>99% purity) was obtained from Ebeikar Tea & Extracts Co., Ltd. (Hangzhou, China). Bovine serum albumin (BSA), reduced glutathione (GSH), 2′,7′-dichlorofluorescin diacetate (DCFH-DA), 5,5′-dithiobis (2-nitrobenzoic acid) (DTNB), nicotinamide adenine dinucleotide phosphate (NADPH), GR (from *Escherichia coli*), sodium selenite, and 1-chloro-2,4-dinitrobenzene (CDNB) were all obtained from Sigma (St. Louis, MO, USA). RIPA reagent, BCA protein assay kit, and caspase 8/9 kits were all purchased from Beyotime Biotechnology (Shanghai, China). Commercial kits for measuring serum SOD, GSH, and GPx levels were purchased from Jiancheng Bioengineering Institute (Nanjing, China). Serum alanine aminotransferase (ALT), aspartate aminotransferase (AST), creatinine (Cr), blood urea nitrogen (BUN), and lactate dehydrogenase (LDH) were measured using a hematology analyzer. ECL Plus reagent and PVDF membrane were products of Bio-Rad Laboratories, Inc. (Hercules, CA, USA). The primary antibodies against β-actin were acquired from Sigma (St. Louis, MO, USA). The primary antibodies against caspase 8, Bax, Bcl-2, anti-rabbit IgG, and anti-mouse IgG secondary antibodies were obtained from Cell Signaling Technology, Inc. (Boston, MA, USA). 

### 3.2. Preparation and Characterization of SeNPs

SeNPs were prepared according to a previously reported method [58,59,60]. Briefly, SeNPs were prepared according to a previously reported procedure with GSH as a reductant, sodium selenite as a selenium source, and BSA as a disperser of SeNPs [58,59,60]. To characterize SeNPs, dynamic light scattering (DLS) (DelsaMax PRO, Beckman, Krefeld, Germany) and transmission electron microscopy (TEM) (HT7700, Hitachi, Tokyo, Japan) were used. The average diameter of SeNPs employed in the present work was 33 nm (Appendix A).

### 3.3. Animals and H22 Model Mice

All animal procedures followed the protocol approved by Jinan University (Guangzhou, China) and the Guidance for the Care of Laboratory Animals of the Ministry of Science and Technology of the People’s Republic of China (2006-398). Male Kunming mice (20–22 g) and animal diets were purchased from Guangdong Provincial Laboratory Animal Center (Guangzhou, China). The mice were housed at a controlled temperature of 24 ± 2 °C, relative humidity of 50 ± 10%, and 12 h light–dark cycle and were allowed free access to food and water ad libitum. 

H22 cells were obtained from Shanghai SLAC Laboratory Animal Co. Ltd. (Shanghai, China), and maintained in our laboratory by propagation in the peritoneal cavity of mice. In brief, each mouse was intraperitoneally injected with a suspension of 20 million viable cells; 48 h later, the highly malignant H22 model mice were established and then used for experiments. The critical parameters of each animal experiment are presented in corresponding figure legends, including the route of administration, experimental period, drug dose, and animal number. 

### 3.4. H22 Cell Collection and Count

H22 cells suspended in the ascitic fluid were collected into tubes and centrifuged at 400× *g* for 5 min at 4 °C. The cells were then rinsed in ice-cold saline and counted in a hemocytometer using the trypan blue dye exclusion method.

### 3.5. ROS Measurement

H22 cells were collected by centrifugation at 400× *g* for 5 min at 4 °C and washed two times with saline according to the same procedure. Finally, each sample was adjusted to 200 μL with 1640 serum-free medium, which contained 20 million viable H22 cells/mL and indicated concentrations of drugs. ROS levels were detected using the aforementioned microplate reader using 488 nm excitation wavelength and 525 nm emission wavelength at 37 °C. Please refer to our previous reports for a detailed scheme [16,18]. 

### 3.6. Determination of Enzyme Activities

To detect extracellular glutathione peroxidase (GPx) or GR activity [61,62], the collected ascitic fluid was centrifuged to remove H22 cells. The supernatants were used to evaluate the total extracellular GPx or GR activity in the peritoneal cavity. The H22 cells were subjected to ultrasonic treatment to detect intracellular enzyme activities. Then the resultant homogenates were centrifuged at 15,000× *g* for 15 min at 4 °C, and the supernatants were used to detect enzyme activities. To detect hepatic enzyme activities [61,63], liver tissues were excised and homogenized in ice-cold conditions. The homogenate was obtained by centrifugation at 15,000× *g* for 15 min at 4 °C, and the supernatants were used to evaluate hepatic TrxR and thioredoxin (Trx) activity levels. Protein levels were determined using the Bradford dye-binding assay with BSA as the standard. GPx and TrxR activities were determined according to the methods of Smith and Levander with some modifications [61,64]. GPx or GR activity was calculated in terms of µmol of NADPH oxidized/min (U). TrxR activity was determined using the NADPH-dependent DTNB reduction method and was presented as µmol of NADPH oxidized/min/mg protein. Trx activity was determined using the method of Holmgren and Björnstedt with rat TrxR1 as a Trx reductase and was presented as µmol of NADPH oxidized/min/mg protein [65]. Caspase 8 and caspase 9 activities were determined following the protocols provided by the manufacturers and were defined as the production of p-nitroaniline (µmol)/min/mg protein at 37 °C. 

### 3.7. Determination of Selenium Level in Tissues

H22 cells were precipitated by centrifugation at 400× *g* for 5 min at 4 °C and sonicated on ice for 5 min with 30 s intervals for testing selenium levels. Selenium levels of tissues and cells were assessed using a 2,3-diaminonaphthalene fluorescence-based method. The fluorescence intensity was excited at 365 nm and recorded at 520 nm, and it was used for calculating selenium content with sodium selenite as a standard.

### 3.8. EGCG Measurement in H22 Cells and Tissues

For detecting the levels of EGCG in the liver and small intestine [66], 0.2 g liver or small intestine was homogenized in 1 mL of the ascorbate-EDTA solution. Then 500 μL homogenate was mixed with 20 μL glucuronidase/sulfatase and incubated at 37 °C for 45 min. After termination by the addition of 200 μL ethanol, the reactions were mixed with 500 μL of methylene chloride, followed by a vigorous vortex. Then the mixture was centrifuged at 16,000× *g* for 5 min at 4 °C. The upper aqueous phase was extracted with ethylacetate ethy and subsequently dried by vacuum centrifugation. The dried solid matter was dissolved in 100 μL of 10% methyl alcohol aqueous solution. After centrifugation at 17,000× *g* for 15 min at 4 °C, the resultant supernatant (5 μL) was analyzed using ultra-high-performance liquid chromatography coupled with a triple quadrupole mass spectrometer (UPLC-QQQ-MS/MS). Chromatographic separation was performed on a Hypersil GOLD column (particle size 1.9 mm; column size 50 × 2.1 mm) with a guard column (particle size 3 mm; column size 10 × 2.1 mm) at 35 °C. The mobile phase consisted of 0.05% aqueous formic acid and (B) methanol. The elution rate was set at 0.3 mL/min, and the gradient of solvent B was as follows according to our previously reported method: 0–1 min, 10%; 1–7 min, from 10% to 30%; 7–7.5 min, from 30% to 70%; then kept at 70% to 8 min; 8–8.5 min, from 70% to 10%; then held at 10% to 10.5 min. The mass spectra were obtained using electrospray ionization in the negative ionization mode. The ion scan range was 100–1000 m/z. The dry gas was set at 6 L/min at 325 °C with a nebulizer gas pressure of 45 psi. In the QQQ-MS/MS experiments, the MRM mode was used to detect the target compounds via selected product ions from the parent ions (EGCG/GCG, 457→169). Analyses of the data were performed using Agilent MassHunter Qualitative Analysis software. One milliliter of H22 cell suspension (100 million cells) was subjected to ultrasonic treatment and used for detecting intracellular EGCG levels with the method mentioned above.

### 3.9. Western Blot Analysis

Total protein concentrations of H22 cells extracted with the RIPA reagent were determined using the BCA protein assay kit. Protein extracts were boiled at 95°C for 10 min in loading buffer and then loaded onto 12% sodium dodecyl sulfate-polyacrylamide gels for electrophoresis. After being separated in polyacrylamide gels, the proteins were transferred onto a PVDF membrane. The membrane was blocked with 5% nonfat dried milk in Tris-buffered saline containing 0.05% Tween 20 (TBS-T) at room temperature (RT). Two hours later, the membrane was incubated with primary antibody diluted according to the dilution ratio provided by manufacturers overnight at 4°C, and incubated with secondary antibody in TBS-T at 2500- or 5000-fold dilution for 60 min after being washed three times with TBS-T at RT. After four washes with TBS-T, antibody bindings in the membrane were detected using the ChemiDoc XRS+ detection system (ECL, Bio-Rad) and quantified by densitometry with the Quantity One Image Analyzer software program (Bio-Rad).

### 3.10. Statistical Analysis

Data are presented as means ± SEM. All statistical analyses were performed using Prism (GraphPad Software, Inc., La Jolla, CA, USA). The differences between groups were evaluated by one-way analysis of variance (ANOVA) with post hoc Tukey or Dunnett test, or two-way ANOVA with post hoc Bonferroni test, as appropriate. The Kaplan–Meier method was used to evaluate survival, and the log-rank test was used to analyze the differences. A *p*-value of less than 0.05 was considered statistically significant. 

## 4. Conclusions

In the current work, we found H22 cells are insensitive to EGCG, as indicated by the highly selective accumulation of EGCG in cancer cells without the generation of a cell proliferation inhibition effect; the underlying mechanisms may involve EGCG-activated extracellular antioxidant enzymes in ascites. In addition, though H22 cells are insensitive to EGCG, the combination of EGCG with SeNPs can protect against SeNP-triggered diarrhea without compromising their suppressing efficacy on PC in mice bearing H22 cells and can reduce SeNP-caused liver toxicity by enhancing antioxidant enzymes in healthy mice (Figure 8). These results provide support for the concept of EGCG being a promising candidate for preventing chemotherapy-induced adverse reactions, including CID and systemic toxicity in cancer patients. The mechanism by which EGCG prevents SeNP-triggered diarrhea in treating PC and the cross-talk between EGCG and selenium in anticancer activity require further evaluation in vivo. 

## Figures and Tables

**Figure 1 molecules-28-03904-f001:**
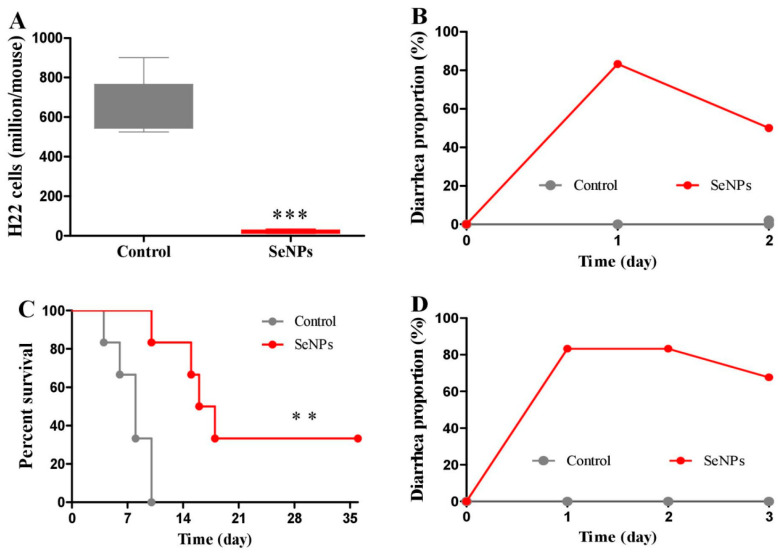
Effects of SeNPs on cancer cell proliferation and survival in mice. Experiment 1: H22 model mice were i.p. injected with 3 mg/kg SeNPs once. At 48 h after the injection, viable cells in the abdominal cavity were collected (*n* = 5). (**A**) Viable cell number. (**B**) Diarrhea rate. Experiment 2: H22 model mice were i.p. injected with 2 mg/kg SeNPs every 4 days for 5 weeks (*n* = 6). (**C**) Survival. (**D**) Diarrhea rate. Data are presented as the mean ± SEM., ** *p* < 0.01, *** *p* < 0.001, compared to control group.

**Figure 2 molecules-28-03904-f002:**
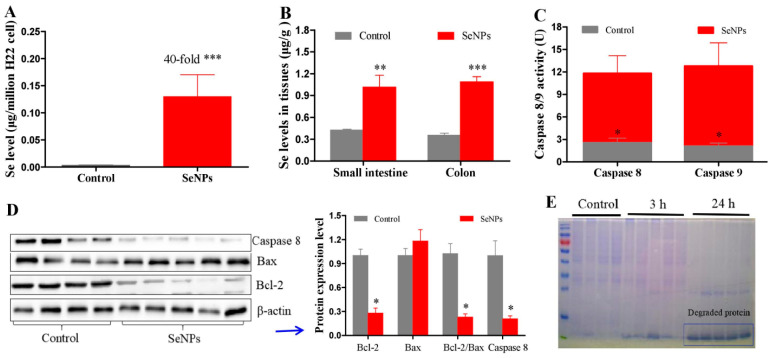
Apoptotic response of H22 cells. Experiment 3: H22 model mice were i.p. injected with 4 mg/kg SeNPs for 3 h or 24 h. (**A**–**D**) Selenium levels, caspase 8 and 9 activities, and apoptosis-associated proteins at 3 h, respectively. (**E**) Coomassie bright blue staining. Data are presented as the mean ± SEM. * *p* < 0.05, ** *p* < 0.01, *** *p* < 0.001, compared to control group.

**Figure 3 molecules-28-03904-f003:**
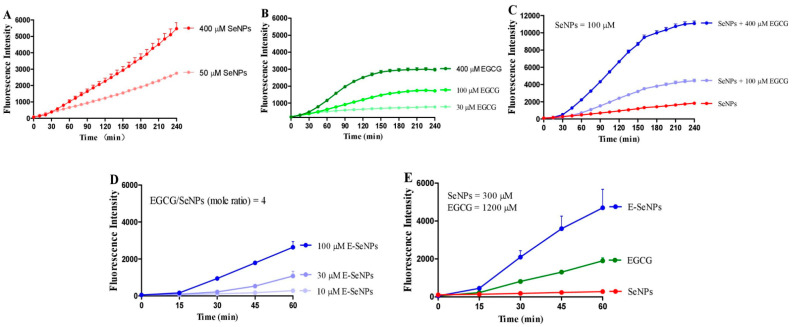
ROS production induced by SeNPs, EGCG, or EGCG combined with SeNPs in H22 cell suspension. (**A**) Dose effect of SeNPs. (**B**) Dose effect of EGCG. (**C**) Dose effect of EGCG on SeNP-induced ROS level. (**D**) Dose effect of E-SeNPs. (**E**) Comparing the effects of SeNPs, EGCG and E-Se on ROS production. Experiments were carried out in cell suspension in the presence of 50 μM DCFH-DA at 37 °C. Data are presented as the mean of two replicates; the error bar represents the range (*n* = 2).

**Figure 4 molecules-28-03904-f004:**
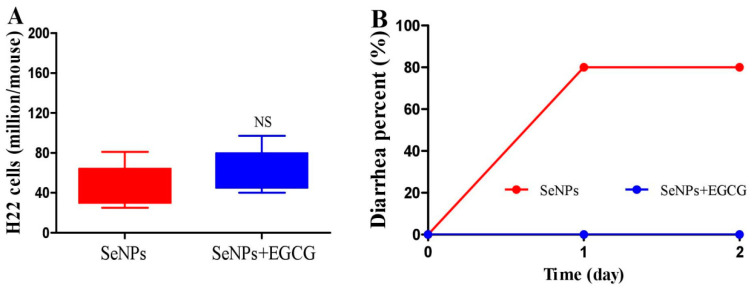
Effect of EGCG on the therapeutic efficacy of SeNPs. Experiment 4: H22 model mice were i.p. injected with SeNPs (4 mg/kg) or EGCG (55 mg/kg) plus SeNPs (4 mg/kg). At 48 h after the injection, viable cells in the abdominal cavity were collected (*n* = 5). (**A**) Cell number. (**B**) Diarrhea rate. Data are presented as the mean ± SEM.

**Figure 5 molecules-28-03904-f005:**
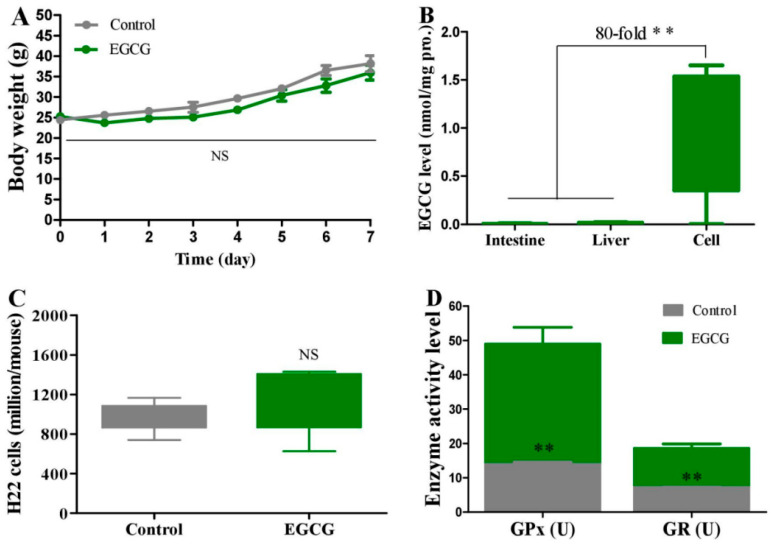
Effect of EGCG on peritoneal carcinomatosis in mice. Experiment 5: Therapeutic efficacy. H22 model mice were i.p. injected with 55 mg/kg EGCG for 7 consecutive days (*n* = 5). (**A**) Body weight. Experiment 6: EGCG levels in different tissues after EGCG treatment. H22 model mice were i.p. injected with 55 mg/kg EGCG; intestine, liver, and cells were collected at 1 h after the injection (*n* = 5). (**B**) EGCG levels. Experiment 7: Preventive effect. Two million viable H22 cells were injected into the peritoneal cavity of mice, and then the mice were i.p. injected with EGCG (50 mg/kg) for 7 consecutive days (*n* = 6). (**C**) Cell number. (**D**) Total activity levels of GPx and GR in ascites in peritoneal cavity. Data are presented as the mean ± SEM. ** *p* < 0.01, compared to control group.

**Figure 6 molecules-28-03904-f006:**
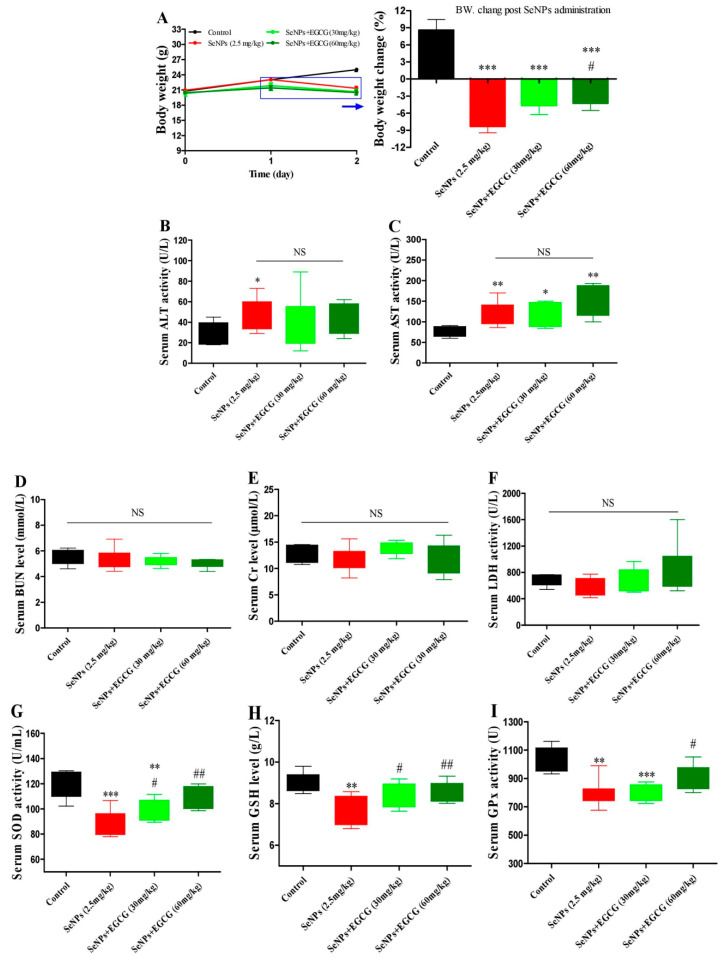
EGCG protects against SeNP-caused body weight loss and oxidative damage in healthy mice. Experiment 8: Healthy mice were i.p. injected with saline as control or EGCG (30 or 60 mg/kg) once; 24 h later, the SeNP and EGCG plus SeNP groups were i.p. injected with 2.5 mg/kg SeNPs (*n* = 6). (**A**) Body weight. (**B**–**F**) Serum ALT, AST, Cr, BUN and LDH levels. (**G**–**I**) Serum SOD, GSH, and GPx levels. Data are presented as the mean ± SEM. * *p* < 0.05, ** *p* < 0.01, *** *p* < 0.001, compared to control group; # *p* < 0.05, ## *p* < 0.01, compared to SeNP group.

**Figure 7 molecules-28-03904-f007:**
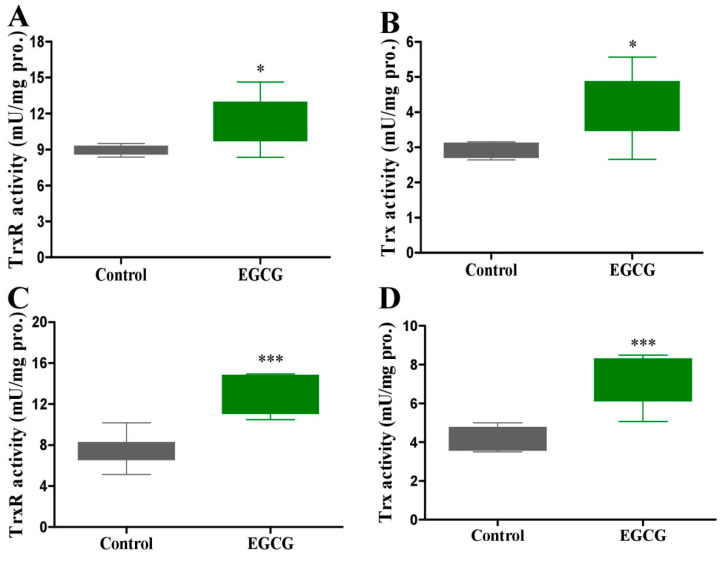
Effect of EGCG on hepatic antioxidant enzymes in healthy mice. Experiment 9: Healthy mice were i.p. injected with 45 mg/kg EGCG for 5 (**A**,**B**) or 7 (**C**,**D**) consecutive days (*n* = 6). TrxR and Trx activities were detected. Data are presented as the mean ± SEM. * *p* < 0.05, *** *p* < 0.001, compared to control group.

**Figure 8 molecules-28-03904-f008:**
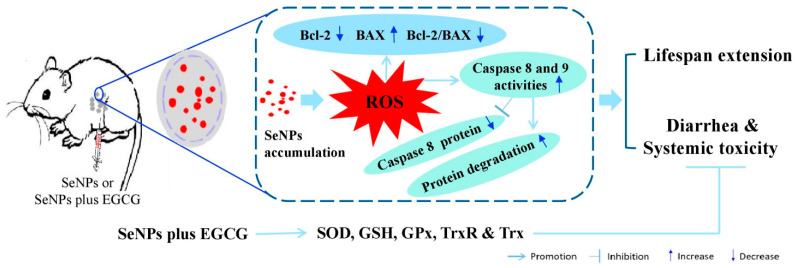
Schematic diagram showing the underlying mechanisms of SeNP anticancer activity and EGCG preventing SeNP-caused adverse reactions.

## Data Availability

The data used to support this study are available from the corresponding author.

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
