# Peer review of "(-)-Epigallocatechin-3-Gallate Attenuates the Adverse Reactions Triggered by Selenium Nanoparticles without Compromising Their Suppressing Effect on Peritoneal Carcinomatosis in Mice Bearing Hepatocarcinoma 22 Cells"

_molecules, 2023, doi:10.3390/molecules28093904_

Round 1

Reviewer 1 Report

Comments and Suggestions for Authors

The present research article entitled “(-)-epigallocatehin-3-gallate attenuates the adverse reactions triggered by selenium nanoparticles without compromising its suppressing effect on peritoneal carcinomatosis in mice bearing hepatocarcinoma 22 cells” contained a interesting work with a good analysis and written way. However, I have some comments as following:

1. Ethical approval number need to be inserted inside manuscript.

2. I n page (5) animal weight range is 20-22 g. However, in Figure 5A calculations start from 25 g??? – please check?

3. In page (14) *** ……ultra-high performance liquid chromatography coupled with triple quadrupole mass spectrometer (UPLC-QQQ-MS/MS). Chromatographic separation was performed on a Hypersil GOLD column (particle size 1.9 mm; column size 50 × 2.1 mm) with a guard column (particle size 3 mm; column size 10 × 2.1 mm) at 35°C. …. ***, I cannot find the results of this experiment.

4. Please add the Standard Deviation for all figures.

5. Methods: Please explain in the text the method of fabricating nanoparticles since the authors used 3 different references.

Author Response

Dear Editor and Reviewers,

Thank you for evaluating the manuscript entitled "(-)-Epigallocatechin-3-gallate attenuates the adverse reactions triggered by selenium nanoparticles without compromising its suppressing effect on peritoneal carcinomatosis in mice bearing hepatocarcinoma 22 cells" (Manuscript ID: molecules-2377531). We have provided point-by-point responses to comments, and revised the manuscript with changes highlighted accordingly. The revised manuscript has been further polished by the authors, including Prof. Daniel Granat, who is currently an editor of Food Chemistry and Food Research International and a guest editor of Current Opinion in Food Science. We appreciate your constructive comments and suggestions, and thank you for your time and serious consideration of this manuscript for publication.

Sincerely yours,

Guangshan Zhao, Ph.D.

Innovation Team of Food Nutrition and Safety Control, College of Food Science & Technology, Henan Agricultural University, Zhengzhou 450002, China

Reviewer 1:

The present research article entitled “(-)-epigallocatehin-3-gallate attenuates the adverse reactions triggered by selenium nanoparticles without compromising its suppressing effect on peritoneal carcinomatosis in mice bearing hepatocarcinoma 22 cells” contained a interesting work with a good analysis and written way. However, I have some comments as following:

  1. Ethical approval number need to be inserted inside manuscript.

Response: Please see the information in page 12.

  1. I n page (5) animal weight range is 20-22 g. However, in Figure 5A calculations start from 25 g??? – please check?

Response: All mice were allowed free access to food and water for 1 week before grouping, thus, the starting weight of the mice was more than 22 g in Figure 5A.

  1. In page (14) *** ……ultra-high performance liquid chromatography coupled with triple quadrupole mass spectrometer (UPLC-QQQ-MS/MS). Chromatographic separation was performed on a Hypersil GOLD column (particle size 1.9 mm; column size 50 × 2.1 mm) with a guard column (particle size 3 mm; column size 10 × 2.1 mm) at 35°C. …. ***, I cannot find the results of this experiment.

Response: UPLC-QQQ-MS/MS method was used to measure the levels of EGCG in H22 cell, intestine and liver, please see Figure 5B.

  1. Please add the Standard Deviation for all figures.

Response: The error bar represents the range or the mean ± SEM in all the figures besides figure 1B-D and Figure 4B. Figure 1B-D and Figure 4B represent the proportion of an indicator in the whole group, so, there was no Standard Deviation.

  1. Methods: Please explain in the text the method of fabricating nanoparticles since the authors used 3 different references.

Response: Thank you for your suggestion. The same method was used in the 3 different references. We revised this sentence (page 11).

Reviewer 2 Report

This manuscript deals with "(-)-Epigallocatechin-3-gallate attenuates the adverse reactions triggered by selenium nanoparticles without compromising its suppressing effect on peritoneal carcinomatosis in mice bearing hepatocarcinoma 22 cells" I suggest a minor correction and require a detailed clarification. A correction should be addressed by the authors as follows: The abstract is not well organized; the sentences are incomplete, and there is no sense of continuity. It would be feasible if you included the significance of the current study in the abstract. A brief description of how the authors selected information from the literature in the databases, as well as what time period they searched for, is missing. The authors should justify and expand the information on the advantages of Epigallocatechin-3-gallate for biomedical applications. Authors should specify the main experimental conditions used based on the evidence from the literature. Where they briefly describe the most important data reported in the literature in a homogeneous manner and reinforce the relevance of Epigallocatechin-3-gallate as novel alternatives. Authors should discuss whether the use of Epigallocatechin-3-gallate nanoparticles  represents a solid alternative to existing therapeutics. Also, please discuss the use of method using green nanomaterials to targeting cells and mitochondria . Please add the below studies to your manuscript in the discussion section and bold your study novelties:

- DOI: 10.1016/j.arabjc.2021.103106

-DOI: 10.7150/ntno.77564

Author Response

Dear Editor and Reviewers,

Thank you for evaluating the manuscript entitled "(-)-Epigallocatechin-3-gallate attenuates the adverse reactions triggered by selenium nanoparticles without compromising its suppressing effect on peritoneal carcinomatosis in mice bearing hepatocarcinoma 22 cells" (Manuscript ID: molecules-2377531). We have provided point-by-point responses to comments, and revised the manuscript with changes highlighted accordingly. The revised manuscript has been further polished by the authors, including Prof. Daniel Granat, who is currently an editor of Food Chemistry and Food Research International and a guest editor of Current Opinion in Food Science. We appreciate your constructive comments and suggestions, and thank you for your time and serious consideration of this manuscript for publication.

Sincerely yours,

Guangshan Zhao, Ph.D.

Innovation Team of Food Nutrition and Safety Control, College of Food Science & Technology, Henan Agricultural University, Zhengzhou 450002, China

Reviewer:

This manuscript deals with "(-)-Epigallocatechin-3-gallate attenuates the adverse reactions triggered by selenium nanoparticles without compromising its suppressing effect on peritoneal carcinomatosis in mice bearing hepatocarcinoma 22 cells" I suggest a minor correction and require a detailed clarification. A correction should be addressed by the authors as follows: (1) The abstract is not well organized; the sentences are incomplete, and there is no sense of continuity. It would be feasible if you included the significance of the current study in the abstract. (2) A brief description of how the authors selected information from the literature in the databases, as well as what time period they searched for, is missing. (3) The authors should justify and expand the information on the advantages of Epigallocatechin-3-gallate for biomedical applications. (4) Authors should specify the main experimental conditions used based on the evidence from the literature. Where they briefly describe the most important data reported in the literature in a homogeneous manner and reinforce the relevance of Epigallocatechin-3-gallate as novel alternatives. (5) Authors should discuss whether the use of Epigallocatechin-3-gallate nanoparticles represents a solid alternative to existing therapeutics. Also, please discuss the use of method using green nanomaterials to targeting cells and mitochondria . Please add the below studies to your manuscript in the discussion section and bold your study novelties:

- DOI: 10.1016/j.arabjc.2021.103106: Cited, reference [2]

-DOI: 10.7150/ntno.77564

Response: (1) Revised (page 2); (2) We mainly collected literature from NCBI published in the latest 30 years about the effect of SeNPs and EGCG on anti-cancer to help us understand the field better. However, the description of literature selection and its time period is optional in a original research paper.

(3) Revised (page 3); (4) Revised (page 11); (5) We added many sentences according to your suggestions in the revised manuscript (page 10).